# The Use of Multimodality Imaging for the Diagnosis of Myocardial Outpouchings and Invaginations: A Systematic Review

**DOI:** 10.3390/life13030650

**Published:** 2023-02-26

**Authors:** Rita Pavasini, Nicola Bianchi, Federica Frascaro, Federico Marchini, Sofia Meossi, Luca Zanarelli, Federico Sanguettoli, Alberto Cossu, Elisabetta Tonet, Giulia Passarini, Gianluca Campo

**Affiliations:** 1UO Cardiologia, Azienda Ospedaliero-Universitaria di Ferrara, 44124 Ferrara, Italy; 2Rita Pavasini, Cardiology Unit, Azienda Ospedaliero Universitaria S. Anna, Via Aldo Moro 8, 44124 Ferrara, Italy; 3University Radiology Unit, University of Ferrara, 44124 Ferrara, Italy

**Keywords:** multimodality imaging, cardiac diverticula, cardiac aneurysms, cardiac crypts, cardiac recesses

## Abstract

Cardiac ventricular outpouchings and invaginations are rare structural abnormalities and usually incidental findings during cardiac imaging. A definitive diagnosis is possible through the use of multimodality imaging. A systematic review of the literature was carried out in November 2022 to identify studies regarding ventricular outpouchings and invaginations. The main aim of the review is to summarize knowledge regarding epidemiology, etiology, diagnosis and prognosis of patients with ventricular outpouchings (aneurisms or diverticula) and invaginations (crypts and recesses). Overall, 26 studies published between 2000 and 2020 were included in the review. Diverticula and congenital aneurysms incidence ranges between 0.6 and 4.1%. Myocardial recesses and crypts range between 9% in the general population and up to 25% in patients with hypertrophic cardiomyopathy. The combined use of echocardiography, cardiac computed tomography (CCT) and cardiac magnetic resonance (CMR) is useful to establish tissue contractility, fibrosis, extension and relationship with adjacent structures for differential diagnosis of both invaginations and outpouchings. In conclusion, both outpouchings and invaginations are rare entities: a definitive diagnosis may be aided by the use of combining multiple imaging techniques, and the treatment depends both on the lesion-specific risk of complications and on the potential association of some lesions with cardiomyopathy.

## 1. Introduction

Cardiac ventricular outpouchings and invaginations are rare structural abnormalities (SA). They are often incidentally diagnosed during cardiac imaging [1,2]. The differentiation between outpouching and invagination is based on the presence of distortion of the outer border of the ventricle: outpouchings are extroversions that extend outside the normal outline of the ventricle wall, whereas invaginations are lesions that involve the myocardium without altering the outer part [3].

The effective prognosis after diagnosis of these features is uncertain, especially because of the lack of systematic studies on large samples of individuals.

Although it is known, for example, that myocardial crypts are associated with hypertrophic cardiomyopathy and recesses with the presence of left ventricular non-compaction, these SA are also frequent in the general population in the absence of any cardiac disease [4,5].

The diagnosis of both outpouchings and invaginations, especially if they are of small dimensions, can be complex and not sufficiently accurate using echocardiography alone, even though this is still the first diagnostic method used. Multimodality imaging with the addition of cardiac computed tomography (CCT) and cardiac magnetic resonance (CMR) to the echocardiographic study alone is the cornerstone to reach a definitive diagnosis [3]. Therefore, the present systematic review aims to summarize the current evidence and the use of multimodality imaging in differential diagnosis and prognostic stratification regarding these rare SA.

## 2. Materials and Methods

We performed a systematic review of literature following the Preferred Reporting Items for Systematic reviews and Meta-Analyses (PRISMA, Appendix A) statement, updated to the 2021 version [6], including studies regarding ventricular diverticula, aneurysms, crypts and recesses. Accordingly, the following terms using medical subject heading (MeSH) strategy were searched: “(ventricular OR cardiac OR myocardial) AND (diverticula OR crypt OR (congenital AND aneurysm) OR recesses)”. The databases analyzed were PubMed, Cochrane Library, BioMed Central and Google Scholar. The literature search was carried out in November 2022. Only full-text articles published in English and in peer-reviewed journals were selected. The following were inclusion criteria: (i) case series (at least 5 patients); or (ii) observational studies on the use of different imaging technique for the diagnosis and treatment of ventricular diverticula, aneurysms, crypts and recesses. Exclusion criteria were (i) articles not regarding humans; (ii) reviews or editorials; (iii) case report or case series on <5 patients. The main aim of the systematic review is to summarize data regarding diagnosis, epidemiology and prognosis regarding ventricular outpouchings and invaginations as assessed by the use of multimodality imaging (namely echocardiography, CMR or CCT). Literature searching, screening of the literature and quality appraisal of selected items were performed by two independent reviewers (NB and FF). Divergences have been solved by discussion and consensus. In case of discordance, a third reviewer (RP) was asked to solve the disagreement and reach consensus. Two reviewers (FM and SM) retrieved data from the included studies. The following information has been collected: regarding the publication (year of publication, journal, country, center of the enrollment, study design), baseline characteristics (age, sex), type of SA analyzed (diverticula, aneurysms, crypts and recesses), type of imaging technique used to reach a diagnosis (echocardiography, CCT, CMR, other), and treatment.

The quality of the included studies was tested using pre-specified electronic forms of MINORS criteria [7].

## 3. Results

Overall, 683 studies were selected. After an evaluation of records, 17 studies published between 2000 and 2020 were included in the review [1,5,8,9,10,11,12,13,14,15,16,17,18,19,20,21,22]. In particular, nine were concerning patients with ventricular outpouchings (diverticula and congenital aneurysms), while eight were concerning patients with ventricular invaginations (crypts and recesses) (Figure 1). Regarding quality assessment, the minimum score obtained for studies with a control group was 18, while the maximum was 22 (Table 1). Otherwise, for studies without a control group, the minimum score obtained was 10, while the maximum was 12 (Table 1). No studies were excluded based on the quality assessment.

Overall, 1998 patients were included in studies regarding ventricular outpouchings and 13,013 in studies regarding ventricular invaginations, of whom 1180 were with confirmed invaginations and 156 with outpouchings, respectively (Table 2).

Data regarding diagnosis of ventricular outpouchings with echocardiography are poor. Epidemiology data are mainly referred to in studies with CCT or CMR made for other reasons. In particular, the prevalence of diverticula range between 0.6% and 4.1% as detected by CCT [8,11,12] and 0.76% with CMR examination. Echocardiography is the first diagnostic tool that can be used for diagnosis, even if in certain locations CCT or CMR offer better anatomical details: CCT provides also coronary artery anatomy to exclude a secondary cause of ventricular aneurysms (VA) and CMR better analyzes the kinesis and fibrosis to differentiate between VA and ventricular diverticula (VD).

The prevalence of crypts/recesses reported in the literature varies according to criteria and methods used to identify them (some authors used a cut-off of 50% of penetration in the myocardium and others 30%) (Table 2) [5]. No studies were excluded from this systematic review based on the criteria used to identify crypts or recesses. Overall, prevalence of crypts detected with CMR is around 6.3–6.7% [5,16] and reaches 9.1% with CT [22]. The identification of crypts is higher in HCM patients or carriers [16,17,18,19,20]. CCT and CMR are the best imaging methods to diagnose crypts [18,19] and recesses.

For both ventricular outpouchings and invaginations, a wide discussion on the main findings of the literature review have been summarized in terms of definitions, epidemiology, etiology, diagnosis, prognosis and treatment in the following paragraph.

## 4. Discussion

### 4.1. Ventricular Outpouchings: Diverticula and Congenital Aneurysms

#### 4.1.1. Definitions

The most frequent ventricular outpouchings are VD and VA. VA are frequently acquired and associated with coronary artery disease and myocardial infarction and rarely are a part of congenital heart disease (in this review, only the congenital ones will be considered). The definitions of VD and VA are not unique in literature [8]: some authors describe VD as ventricular outpouchings with contractile activity synchronal with the ventricle and with all three myocardial layers in its wall; conversely, VA is defined as a ventricular outpouching with akinetic or dyskinetic movement and fibrotic wall not presenting all myocardial layers [9]. Other authors divide VD into muscular or fibrous [23] and differentiate them from VA by the width of the communication within outpouching and ventricular cavity: VA have larger communication than VD [10].

#### 4.1.2. Epidemiology

The epidemiology of these ventricular lesions is debated [24] and in part unknown seeing the rarity of their incidence in the general population. Right VD are even rarer then left VD. Right VD are described only in case series and frequently they are associated with left VD [1,9].

In previous series of echocardiographic, autoptic, surgical or cardiac catheterization studies on the prevalence of cardiac outpouchings were very few (0.04–0.26%) [11]. Some authors analyzed CCT or CMR exams utilized for other reasons to investigate the incidence of these rare, but even underdiagnosed, SA.

De Bruecker et al. analyzed 482 CCTs [12] and found 20 left VD (4.1% of patients) with a dimension ranging between 0.5 and 1.4 cm; the most frequent location was in the inferior septal left ventricular wall and 30% of patients had multiple diverticula.

Nakazono et al. [11] analyzed 324 patients that underwent 256-slice multidetector CCT and found 18 left VD in 11 patients (3.4% of patients) and 3 right VD in 2 patients (0.6% of patients). In both patients with right VD, left VD were also present.

Srichai et al. [8] screened 680 CCT and 23 VD were found in 15 patients (2.2% of patients) with no case of right VD. All VD were located in the inferior and infero-septal wall of the left ventricle [8].

Another study used CMR to define the prevalence of VD: Aquaro et al. [13] reviewed 3273 CMR, excluding patients with a history of cardiac disease, non-cardiac congenital disease or pathological findings at CMR. Isolated diverticula were found in 25 patients (0.76%) with no apparent cardiac disease.

In addition, a small case series [14] described pseudoaneurysms of the left ventricular outflow tract (LVOT) with a mean diameter of 26 mm and with a compression effect on adjacent structures. In another case series, congenital sub-valvular left ventricular aneurysms were present in the case of congenital weakness of fibrous annuli and were often associated with an aneurysm of the sinus of Valsalva [15]. Interestingly, in the Chinese literature, VD are described in adult patients, with a higher predominance for the male sex and resulting in less complications than in the Western literature [25].

Ohlow et al. reviewed 809 cases of left VD and VA and found that VA were larger and more frequently in the sub-mitral location; VD, instead, were localized in the left ventricular apex and were more frequently associated with cardiac or extracardiac anomalies [26].

#### 4.1.3. Etiology

The etiology of VD and VA is controversial; the most accepted hypothesis [9] is that apical VD are part of Cantrell syndrome in which a lot of defects with midline thoracoabdominal formation in embryologic development (such as omphalocele, anterior diaphragmatic hernia, ectopia cordis, tetralogy of Fallot and pericardial defects) are included. In this case, the VD formation mechanism may result from an abnormal fusion between the cardiac loop and the yolk sac before it descends (the so called “traction theory” [26]), leading to the formation of muscular apical VD.

The etiology of non-apical VD and VA [9] has been attributed to a focal defect of the muscular ventricular wall due to an intrinsic abnormality in embryogenesis. VA have also been suspected to be acquired in the prenatal period, potentially because of a viral infection or coronary lesions including stenosis, hypoplasia, and localized intimal proliferation.

In right VD, the etiology [1] of apical type is not associated with Cantrell syndrome and right VD are usually associated with pericardial effusion without cardiac failure. Non-apical right VD originate preferentially from the infundibulum or from the anterosuperior aspect of the right ventricle. Large anterosuperior right VD are rather broad-based accessory chambers and are frequently associated with congenital cardiac malformations. The spectrum of commonly associated defects consists of peri-membranous and muscular ventricular septal defects, tetralogy of Fallot, and double-outlet RV.

#### 4.1.4. Diagnosis

For the diagnosis of left ventricular outpouchings, echocardiography should be the initial diagnostic tool because it is inexpensive, reproducible and demonstrates localization, relation to adjacent structures and eventual associated congenital heart abnormalities [23] (Figure 2 and Figure 3). However, both VD and VA are difficult to see in a certain location [1]; therefore, in growing children or in adult echocardiography may not be sufficient as diagnostic tools for VD or VA, and CMR or CCT can be other second level diagnostic methods [8,9] (Figure 2 and Figure 3). Typical findings in CCT for VD are protruding structures with a sac-like or tube-like shape and narrow orifice from the ventricular lumen. Typically, the differential diagnosis with trabeculation is made with the lesion depth of at least half of the ventricular wall or >5 mm in the right ventricular wall [11]. The utility of evaluation of VA with CCT is more debated since it can provide information on coronary artery anatomy in order to exclude secondary causes of VA. CCT should be used for evaluation of morphology and tissue characteristics only when echocardiography is not conclusive and CMR is unavailable [27].

When an alteration of the myocardial wall is seen with echocardiography, cardiac computer tomography (CCT) or cardiac magnetic resonance (CMR) should be used to differentiate between invaginations or outpouchings. When the abnormal structure involves the outer profile of the myocardial wall, it is defined ventricular outpouching, otherwise ventricular invagination. When a ventricular outpouching is diagnosed, if the structure is akinetic or dyskinetic with a fibrotic wall and a large neck, it is called an aneurysm, and coronary artery anatomy must be investigated and a CMR performed. If the outpouching has synchronous movements with the ventricular wall, with a narrow neck, and the wall of the lesion contains all myocardial layers, it is is called diverticulum and CCT or CMR must be performed to investigate eventual associated congenital cardiac abnormalities or compression of adjacent structures. When a ventricular invagination is diagnosed, if it involves more than 50% of myocardial wall thickness, it is named a crypt and CMR must be performed to investigate the hypertrophic cardiomyopathy phenotype. Otherwise, if the invagination involves less than 50% of the myocardial wall thickness, it is named recess and CMR must be performed to investigate the left ventricular non-compaction phenotype.

CMR is another diagnostic method very accurate in diagnosis, follow-up and differential diagnosis of VD and VA, with the identification of a fibrous wall of VA with late gadolinium enhancement (LGE) [9]. In CMR, VA are seen as outpouchings with akinetic or dyskinetic movement, large neck, and with fibrotic wall identified with LGE sequences, as opposite VD are characterized by synchronous contraction with the ventricular wall, narrow neck and an absence of fibrosis in LGE sequences. CMR can also be useful in diagnosis of eventually associated congenital cardiac abnormalities [9].

McMahon et al. [10] analyzed CMR images of 25 cases of VA and one of VD with a clear definition of morphology of VA and its functional implications. In this study, VAs located in the apex or free wall were significantly larger (24 ± 29 vs. 3 ± 2 mL/m^2^, *p* = 0.02) and more frequently associated with late enhancement. VA volume, but not location, correlated with left ventricular size (*p* < 0.0001) and ejection fraction (*p* < 0.0001).

#### 4.1.5. Prognosis and Treatment

In clinical practice, most VD are asymptomatic and with good long-term prognosis [12], but some arrhythmias such as ventricular tachycardia and atrial fibrillation can be associated with VD. VD can cause rare severe complications such as heart failure, systemic embolization, valvular regurgitation, ventricular rupture and sudden death, and these complications are more frequent seen in children [11].

Left VA more frequently presents with ventricular arrhythmias than VD [26] and outpouchings localized in the septal wall are associated with a higher prevalence of embolic and arrhythmic complications [13], but ventricular rupture, syncope and embolic events are similar between VA and VD [26].

VA have worse prognosis than VD (mean survival 30% at 4 years vs. 80% at 10 years *p* < 0.00001) [9,26]. Cardiac death occurs in VA patients more frequently for congestive heart failure and in VD patients the first cause of death is tamponade due to ventricular wall rupture [26].

Patients with lateral VD have a significant lower ejection fraction than those with VD in other locations [13]: an explanation for this phenomenon may be that lateral VD represent a focal variant of left ventricular non-compaction with a single deep recess associated with a progressive impairment of the systolic function, but further studies are necessary to confirm this hypothesis [11].

The treatment of symptomatic VD, and symptomatic or asymptomatic VA (taken in consideration the worse mid-long-term prognosis) is often surgical. In asymptomatic VD, the indication for a surgical procedure is debated because of the risk of serious adverse events such as embolism, arrhythmias, cardiac failure and rupture but with a low incidence. In this case, small studies showed that a conservative strategy might be chosen with good mid-term prognosis [9,11]. An exception are children in whom risk of ventricular rupture is higher, and, in this case, surgical treatment is often recommended [11].

### 4.2. Ventricular Invaginations: Crypts and Recesses

#### 4.2.1. Definitions

Crypts and recesses are defined as narrow blood-filled invaginations of the left ventricular (LV) wall that do not extend beyond the myocardial margin. Both contract partially or completely during systole due to surrounding tissue but differentiate because crypts penetrate more than 50% of the myocardial thickness and for recesses, conversely, less than 50% [3].

#### 4.2.2. Epidemiology

Child et al. reported a prevalence of 6.3% in a large population of 1020 CMR studies [5]. The same prevalence was found in the analysis of Petryka et al. [16] in a retrospective study. Crypts were found more frequently in patients with hypertrophic cardiomyopathy (HCM) (15.6%), myocarditis (15.4%) and arterial hypertension (13.6%) (*p* = 0.12). Fewer cases were identified in individuals with HCM family history-negative phenotype (5%) and without structural cardiac abnormalities (8.7%) [16].

Arow et al. [17] analyzed a large cohort (n = 393) using contrast-enhanced computed tomography (CT) and found that the overall prevalence of crypts was 9.4%. As previously described, crypts were significantly more prevalent in HCM patients (24.7%), where they appeared longer, more numerous, with a larger area and more likely to be in the infero-basal segment.

Other studies identified a higher number of infero-septal crypts (61–81%) in HCM mutation patients without LV hypertrophy (carriers) using CMR. To the contrary, crypts were uncommon in patients with LV hypertrophy, deducing that they could disappear with the LV thickening [18,19]. Interestingly, German et al. noted that specific imaging planes focalized on the infero-septum increased the sensitivity to visualize crypts in carrier individuals [18]. As shown by Brouwer et al. [20], only half of the crypts visualized at a modified two-chamber view appeared also on standard long-axis cine images, and the detection of two or more invaginations has a 100% positive predictive value to identify carriers.

#### 4.2.3. Etiology

Although the etiopathogenesis is not clear, it is believed that they represent a failure to resorb the trabeculated myocardium during normal embryogenesis. These features remained unrecognized for long time and were described for the first time in 1958 in autoptic hearts of adolescents who died unexpectedly. Progressively, myocardial crypts are found to be in neat myocardial disarray, principally in the infero-septum in individuals with HCM [21], supporting the idea that the disruption of myocyte architecture is a morphological substrate for crypt formation [2,3].

Isolated crypts were considered for a long time to be a morphologic marker of HCM; however, advanced imaging techniques allowed them to also be identified in other pathologies and in healthy individuals [5].

#### 4.2.4. Diagnosis

As demonstrated in different studies, two-dimensional echocardiography is not able to detect such small structures; to the contrary, CMR and CT are the best cardiac modalities to make a diagnosis of crypts (Figure 2 and Figure 3) [18,19]. CMR allows us to assess the invagination’s disappearance through the myocardial wall during the diastolic phase and thus differentiate between crypts and recesses. Furthermore, both entities are characterized by normal signal intensity because fibrosis is rarely present in patients with sarcomeric mutation and without expressed hypertrophy. In Germans et al.’s study, areas of focal fibrosis were identified only in carrier patients and were limited to hypertrophied myocardial areas [18].

ECG-gated CCT has high spatial and temporal resolution and permits the evaluation of the myocardial wall [28]. The study of regional wall contractility is possible only with ECG-gated CCT angiography, but the exposure to a high radiation dose raises concerns [28].

Prominent trabeculations on the luminal surface and recesses that extend into the myocardial wall may be representative of a pathology defined as left ventricular non compaction (LVNC), a genetic cardiomyopathy transmitted as an autosomal dominant trait. The pathogenesis is like that of crypt formation, and it is related to a defect of embryonic myocardial morphogenesis. There have been two forms described: the isolated LVNC, which occurs in absence of other conditions, and the non-isolated one, when it is associated with other congenital heart diseases [2,4].

Contrary to crypt identification, the first suspect of LVNC originates from echocardiographic images and the diagnosis is based on the following criteria: presences of numerous trabeculations especially in the mid-ventricular and apical segments; two-layer myocardium with a non-compacted (N) and compacted (C) ratio (N/C) > 2 in the adults and >1.4 in children; and perfusion of intertrabecular spaces demonstrated by color Doppler [3,4].

CMR has stricter criteria than echocardiography and requires a NC/C ratio > 2.3 in the end-diastolic phase to make a LVNC diagnosis. Furthermore, the presence of increased signal intensity related to the blood flow between the myocardial trabeculae is attributable to LVNC. Subendocardial LGE can be also detected [29].

Like other techniques described, CT is also able to analyze accurately the two layers (compacted and non-compacted) of the LVNC. Melendez-Ramirez et al. proposed a NC/C ratio > 2.2 in at least two segments to diagnose LVNC accurately. CT has the disadvantage of using iodinated contrast medium and is characterized with low sensitivity for the fibrous tissue but can be a good choice for patients with an implanted device or those requiring a coronary artery evaluation [30,31].

The LVNC has always been considered a rare entity with an adult prevalence < 0.3%; however, studies have demonstrated that echocardiographic criteria can overestimate the LVNC diagnosis, especially in in black individuals and patients with LV dysfunction [4]. Contrast-echocardiography has a higher sensitivity to detect LV recesses, but CMR is superior to 2D-echocardiography, and it is used to confirm or rule-out LVNC diagnosis and to differentiate physiological hyper-trabeculation in athletes [3,4].

#### 4.2.5. Prognosis and Treatment

The prognostic role of crypts was investigated in two major studies and neither recorded an increased risk of major adverse cardiovascular events (MACE) in the affected population compared to the general one [19,22]. Furthermore, no morphological pattern of crypts was associated with an increased hazard ratio of MACE [19,22]. In Maron et al.’s study [19], after a follow up period of 1.5 ± 1.7 years, each patient with crypts was alive and none of them had developed heart failure symptoms or underwent surgical procedures (myectomy or alcohol ablation) or reported an appropriate implantable cardioverter defibrillator (ICD) intervention.

The high prevalence of crypts in patients with a family history of HCM may represent a marker of sarcomeric mutations—carrier status—before a manifested hypertrophy, and it could be used to select patients that deserve a genetic test or a closer echocardiographic follow-up. In other clinical scenarios, crypt identification does not have clinical significance and does not require additional investigations [16].

Finally, the clinical history of patient with crypts and LVNC varies radically because the first ones have a preserved LV ejection fraction and are generally asymptomatic; the other ones develop a progressive reduction in cardiac function, heart failure (HF) symptoms, arrhythmias and thromboembolic events that might require HF treatments, anticoagulation and ICD implantation [20].

## 5. Conclusions

Outpouchings and invaginations are a more frequent incidental finding than previously thought due to the increasing use of CCT and CMR (Figure 4). Diverticula and congenital aneurysms are outpouchings, rare entities whose definition, epidemiology and prognosis remain uncertain. In their diagnosis, the combined use of echocardiography, CCT and CMR is useful to establish tissue contractility, fibrosis, extension and a relationship with adjacent structures. The therapy of outpouchings might not be generalized, and an individualized risk stratification of the patient must be performed prior to any therapeutic approach. Invaginations include myocardial recesses and crypts. Additionally, in this case, multimodal imaging with echocardiography in association with higher spatial resolution methods (such as CCT and CMR) has allowed the identification of them also in the general population, and in patients with other cardiac diseases, such as LVNC or HCM mutation carriers, for whom whether cardiomyopathy is diagnosed or not determines treatment and prognosis. In conclusion, in cases of suggestive clinical symptoms or in patients with a family history of cardiomyopathy (e.g., HCM or LVNC), CMR or CCT could be used in addition to echocardiography for the detection of outpouchings and invaginations. In the general population, multimodality imaging could be used in the case of echocardiographic doubt of SA, trying to prefer CMR to CCT.

## Figures and Tables

**Figure 1 life-13-00650-f001:**
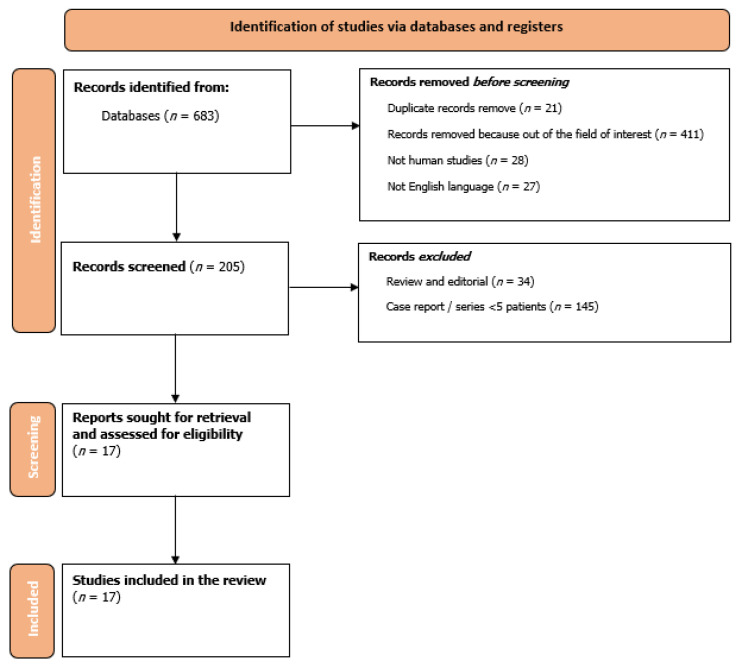
Flow chart of review.

**Figure 2 life-13-00650-f002:**
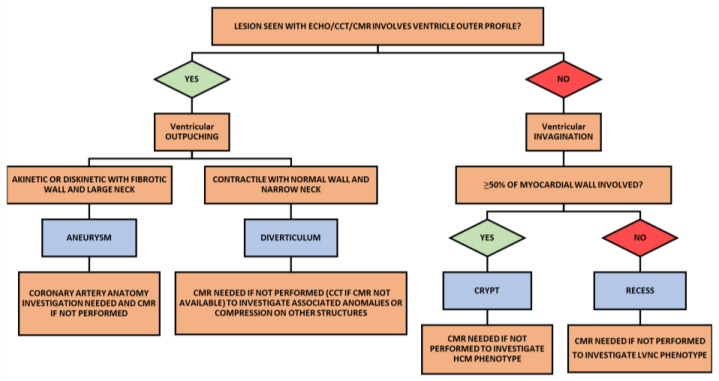
Diagnostic algorithm outpouchings/invaginations.

**Figure 3 life-13-00650-f003:**
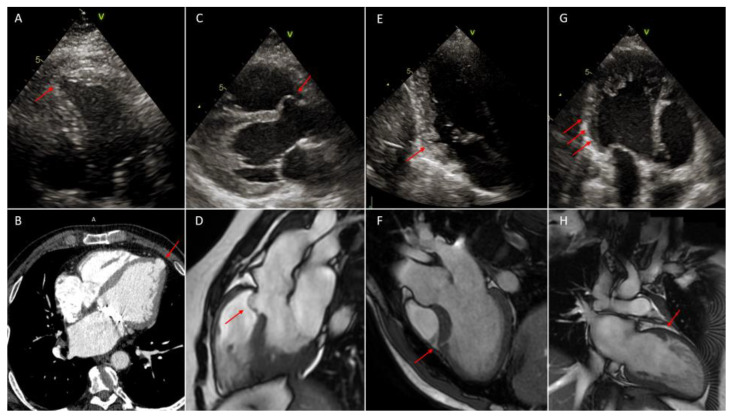
In all figures red arrows indicate the lesion. (**A**): left ventricular apical diverticulum, 2D-Multimodality imaging of outpouchings/invaginations. Echocardiography in apical four chambers view; (**B**): left ventricular apical diverticulum, cardiac contrast-enhanced computer tomography (CCT); (**C**): left ventricular congenital septal aneurysm in left ventricular non-compaction (LVNC), 2D-echocardiography in parasternal long axis (PLAX) view; (**D**): left ventricular congenital septal aneurysm, cardiac magnetic resonance (CMR); (**E**): inferior basal myocardial crypt; 2D-echocardiography in apical two chambers view; (**F**): myocardial crypt of the mid antero-septal wall in hypertrophic phenotype of basal septum, CMR; (**G**): multiple myocardial recesses in (LVNC), 2D-echocardiography in apical parasternal long axis (APLAX) view; (**H**): myocardial recess in left ventricular non-compaction (LVNC), CMR.

**Figure 4 life-13-00650-f004:**
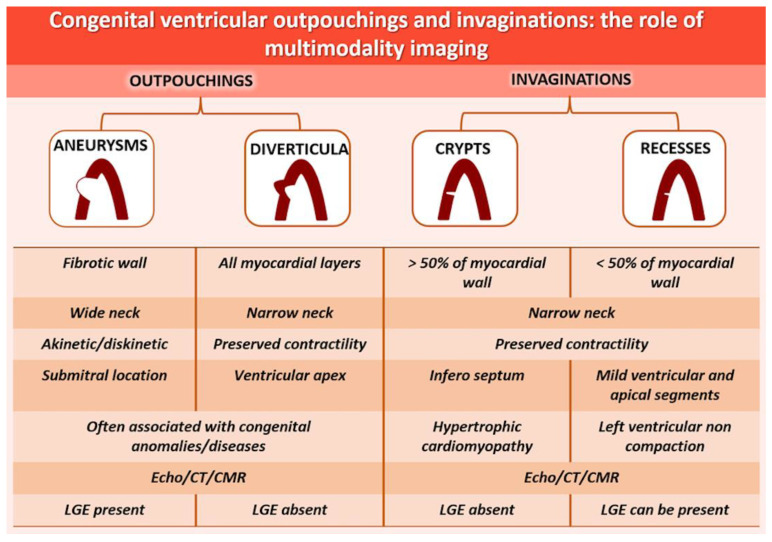
Key elements for differential diagnosis of ventricular outpouchings and invaginations. Echo: echocardiography; CT: computer tomography; CMR: cardiac magnetic resonance; LGE: late gadolinium enhancement.

**Table 1 life-13-00650-t001:** Quality assessment of studies included in the systematic review using MINORS criteria. For the following items, 0 points were given if “not reported”; 1 point if “reported but inadequate” and 2 points if “reported and adequate”. Q1: A clearly stated aim; Q2: Inclusion of consecutive patients; Q3: Prospective collection of data; Q4: Endpoints appropriate to the aim of the study; Q5: Unbiased assessment of the study endpoint; Q6: Follow-up period appropriate to the aim of the study; Q7: Loss to follow up less than 5%; Q8: Prospective calculation of the study size; Q9: An adequate control group; Q10: Contemporary groups; Q11: Baseline equivalence of groups; Q12: Adequate statistical analyses. Q8–Q12 used only for studies with control group.

References	Q1	Q2	Q3	Q4	Q5	Q6	Q7	Q8	Q9	Q10	Q11	Q12	Overall
Wiegand et al. 2014 [1]	2	2	0	2	2	2	2	0	0	0	0	0	12
Child et al. 2014 [5]	2	2	0	2	2	2	2	0	2	2	2	2	20
Srichai et al. 2006 [8]	2	0	0	2	2	2	2	0	2	2	2	2	18
Marijon et al. 2006 [9]	2	2	0	2	2	2	2	0	0	0	0	0	12
McMahon et al. 2007 [10]	2	0	0	2	2	2	2	0	0	0	0	0	10
Nakazono et al. 2012 [11]	2	2	0	2	2	2	2	0	2	2	2	0	18
De Bruecker et al. 2011 [12]	2	2	0	2	2	2	2	0	0	0	0	0	12
Aquaro et al. 2013 [13]	2	2	0	2	2	2	2	0	2	2	2	2	20
Gelehrter et al. 2002 [14]	2	2	0	2	2	2	2	0	0	0	0	0	12
Deshpande et al. 2000 [15]	2	2	0	2	2	2	2	0	2	2	2	2	20
Petryka et al. 2014 [16]	2	2	0	2	2	2	2	0	2	2	2	2	20
Arow et al. 2019 [17]	2	2	0	2	2	2	2	0	2	2	2	2	20
Germans et al. 2006 [18]	2	2	0	2	2	2	2	0	0	0	0	0	12
Maron et al. 2012 [19]	2	2	2	2	2	2	2	0	2	2	2	2	22
Brouwer et al. 2012 [20]	2	2	0	2	2	2	2	0	2	2	2	2	20
Urbano-Moral et al. 2019 [21]	2	2	2	2	2	2	2	0	2	2	2	2	22
Sigvardsen et al. 2020 [22]	2	2	2	2	2	2	2	0	2	2	2	2	22

**Table 2 life-13-00650-t002:** Key message of each study included in review. SA: structural abnormality; CMR: cardiac magnetic resonance; CT: computed tomography; HCM: hypertrophic cardiomyopathy; LVOT: left ventricle outflow tract; ICM: ischemic cardiomyopathy; NICM: non-ischemic cardiomyopathy; LV: left ventricular; NS: not specified.

References	Study Type	N	Male n (%)	SA Analyzed	Patients with Confirmed SA	SA Definition	Modality Imaging Used	Outcome
Wiegand et al., 2014 [1]	Retrospective	5	Unknown	5	5	Diverticula: Ventricular outpouching with synchronous contractions during cardiac systole.	Echocardiography	The rare antero-superior right ventricular diverticula appear to be a specific congenital cardiovascular anomaly and are frequently associated with other congenital defects.
Child et al., 2014 [5]	prospective	1020	617 (61)	64	64	Crypts: invaginations > 50% of the myocardial thickness	CMR	The overall prevalence of crypts was 6,3%. No significant differences were found between NICM and ICM patients. Crypts were more prevalent in patients referred for family screening (23%, *p* < 0.001)
Srichai et al., 2006 [8]	Retrospective	680	Unknown	15	15	Diverticula: Ventricular outpouching with involvement of at least half of the compacted myocardial wall thickness in the diastolic phase.	CT	Prevalence of diverticula in CT performed for other reasons was 2.2%. All patients had normal left ventricular function.
Marijon et al., 2006 [9]	Retrospective	22	11 (50)	22	22	Diverticula: ventricular outpouching with synchronous contractions during cardiac systole and muscular fibers on histological examination. Aneurysms: large akinetic or dyskinetic outpouchings with wide connection to the ventricle and endocardial or transmural fibrosis on histologic examination	Echocardiography, CMR	Diverticula and aneurysms are different pathologies and should not be confused. Aneurysms have poorer prognosis compared to diverticula (*p* < 0.0001).
McMahon et al., 2007 [10]	Retrospective	23	16 (70)	26	26	Aneurysms: segment of the left ventricle wall with a thin-walled outpouching with a wide communication with the cavity. Diverticula: ventricular outpouchings with narrow communication with ventricular chamber	CMR	Apical and free-wall aneurysms were larger (*p* = 0.02) and more frequently associated with scar tissue (*p* = 0.02). Aneurysm volume was associated with left ventricular size (*p* < 0.0001) and EF (*p* < 0.0001).
Nakazono et al., 2012 [11]	Prospective and Retrospective	324	188 (58)	18	18	Diverticula: Protruding structure with a sac-like or tube-like shape and narrow orifice with lesion depth at least half of the compacted myocardial wall.	CT	Incidence of right and left ventricular diverticula was 0.6% and 3.4%, respectively, in 256-slice CT performed for other reasons.
De Bruecker et al., 2011 [12]	Retrospective	542	Unknown	20	20	Diverticula: Ventricular outpouching with involvement of at least half of the compacted myocardial wall thickness in the diastolic phase.	CT	Incidence of left ventricular diverticula was 4.1% in dual source CT performed for other reasons.
Aquaro et al., 2013 [13]	Retrospective	377	Unknown	25	25	NS	CMR	Prevalence of ventricular diverticula in patients undergoing CMR for other reasons and without other cardiac pathologies was 0.76%.
Gelehrter et al., 2002 [14]	Retrospective	9	Unknown	9	9	NS	Echocardiography	LVOT pseudoaneurysm is rare and can impinge upon other structures and is frequently located in the area of fibrous continuity between mitral and aortic valve.
Deshpande et al., 2000 [15]	Retrospective	16	11 (69)	16	16	NS	Echocardiography Autoptic	Subvalvular left ventricular aneurysm are rare entities often associated with other congenital cardiac disease.
Petryka et al., 2014 [16]	Retrospective	686	377 (55)	46	46	Crypts: invaginations penetrating > 50% of the myocardial thickness	CMR	Overall prevalence of crypts was 6.7% with a higher detection in HCM patients (16%). Crypts were identified in patients without CMR anomalies (8.7%).
Arow et al., 2019 [17]	Prospective	393	239 (61)	37	37	Crypts: invaginations penetrating > 50% of the compact myocardium	CT	Myocardial crypts were more prevalent in HCM patients (24.7%) with a larger area and deeper penetration.
Germans et al., 2006 [18]	Prospective	32	12 (38)	13	13	NS	Echocardiography, CMR	Crypts were identified in the 81% of carriers (13/16) by CMR in the inferoseptal LV wall and they were not observed in healthy volunteers. 2D-echo was not able to detect crypts in any case.
Maron et al., 2012 [19]	Prospective	390	254 (85)	29	29	Crypts: LV invaginations extending by visual assessment ≥ 50% of wall thickness but not fully penetrant and not visible at end-systole	Echocardiography, CMR	The crypts’ prevalence was higher in carriers than in HCM group (61% vs. 4%, *p* < 0.001). Any crypts were detected in the control group and by 2D-echo.
Brouwer et al., 2012 [20]	Prospective	295	160 (54)	61	61	Crypts: disruption of compacted myocardium penetrating the LV wall of ≥30% and showing total or subtotal obliteration during systole	CMR	Carrier’s group had a higher number of crypts (*p* < 0.001) and a higher penetration in the myocardium (*p* < 0.01). Modified two chamber view was more sensitive for crypt visualization then standard long axis. Detecting ≥ 2 crypts have a 100% positive predictive value to identify carriers in family screening.
Urbano-Moral et al., 2019 [21]	Prospective	100	65 (65)	15	15	Crypts: sharp-edged disruptions penetrating ≥ 30% within the compact myocardium	Standard and contrast echocardiography, CMR	Contrast echocardiography (n = 94) identified crypts in 15 patients (16%) with HCM and appeared non inferior to CMR.
Sigvardsen et al., 2020 [22]	Prospective	10097	4411 (44)	915	915	Crypts: narrow blood-filled invaginations from LV cavity extending > 50% of the compacted myocardial wall by visual inspection	CT	Crypts were detected in 9.1% of patients and more frequently in the general population. During the follow-up (median 4 years), crypts’ detection was not associated with an increased hazard ratio of major adverse cardiovascular events.

## Data Availability

No new data were created or analyzed in this study. Data sharing is not applicable to this article.

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
