# Peer review of "The Use of Multimodality Imaging for the Diagnosis of Myocardial Outpouchings and Invaginations: A Systematic Review"

_life, 2023, doi:10.3390/life13030650_

Round 1

Reviewer 1 Report

The comprehensive review by Pavasini et al. summarizes the current knowledge about ventricular abnormalities. I find this work of interest, which can be also a good database. I have some minor remarks (to be considered).

In my opinion, the many parts of the discussion should be transferred to the results section -Result section lacks the summary of findings. It would be also appreciated to include more concluding remarks for each aspect.

Please have a look at the grammar and typos. for example: "aims to summaries"

Author Response

Reviewer #1

Comment #1

The comprehensive review by Pavasini et al. summarizes the current knowledge about ventricular abnormalities. I find this work of interest, which can be also a good database. I have some minor remarks (to be considered).

In my opinion, the many parts of the discussion should be transferred to the results section -Result section lacks the summary of findings. It would be also appreciated to include more concluding remarks for each aspect.

Reply # 1

We thank the Reviewer for this comment. We have better summarized Results. We also added Figure 4 to better summarize concluding remarks for a diagnosis of each SA.

Modified text: Section Results

Data regarding diagnosis of ventricular outpouchings with echocardiography are poor. Epidemiology data are mainly referred to study with CCT or CMR made for other reasons. In particular prevalence of diverticula range between 0.6% and 4.1% as detected by CT [8;11-12] and 0.76% with CMR examination. Echocardiography is the first diagnostic tool that can be used for diagnosis, even if in certain locations CT or CMR offer better anatomical details: CCT provides also coronary artery anatomy to exclude secondary cause of VA and CMR better analyses the kinesis and fibrosis to differentiate between VA and VD.

The prevalence of crypts/recesses reported in the literature varies according to criteria and methods used to identify them (some authors used a cut-off of 50% of penetration in the myocardium and other the 30%) (Table 2)[5]. No studies were excluded by this systematic review based on the criteria used to identify crypts or recesses. Overall, prevalence of crypts detected with CMR is around 6.3-6.7% [5; 16] and reach 9.1% with CT [22]. Identification of crypts is higher in HCM patients or carriers [16-20]. CT and CMR are the best cardiac modalities to make diagnosis of crypts [18-19] and recesses.

Modified text: Figure 4

Comment # 2

Please have a look at the grammar and typos. for example: "aims to summaries"

Reply #2

We are sorry for typos. We have extensively reviewed the manuscript and amended typos.

Modified text: Section “simple summary”

aims to summarize

Reviewer 2 Report

the paper can be published after correction of the minor spelling mistakes.

Author Response

Reviewer #2

Comment # 1

The paper can be published after correction of the minor spelling mistakes.

Reply #1

We really thank the Reviewer # 2 for appreciating our work. We extensively revised English and Grammar in the revised version. Thank you.

Reviewer 3 Report

 In this systematic review, Pavasini et al. examine studies on various imaging techniques for ventricular outpouchings (diverticula and congenital aneurysms) and invaginations (recesses and crypts). Based on the more frequent occurrence of such imaging findings with CT/MRI, they propose the use of the latter on top of echocardiography for outpouching/invagination workup.

The objective/aim of this review is not clear, especially given the recent systematic review by Hojland et al. https://doi.org/10.1002/jcu.23155 As far as I can understand, the authors aim to summarize the data on outpouchings/invaginations from CT/MRI studies.

As ventricular diverticulum definition is not universally accepted, for every study including such findings, the definition that the authors applied should be visible.

When specific case series/cohorts are mentioned (for example lines 23-50), the ancestry-race of subjects should be included (caucasians, asians, africans…) as it may affect the findings.

The authors conclude that outpouchings/invaginations are more frequently detected with CT/MRI than with echo, nevertheless they recommend echo as the first-line imaging modality. On the other hand, given the usually benign course of such findings, it is debatable both if a negative echo of adequate quality should lead to MRI and if a positive echo (for example indicative of one or two LV recesses or a typical aneurysm) should be followed by an MRI scan. Indications for cardiac CT should be restricted to non-availability of MRI. Thus, the need for additional imaging (on top of TTE) should be judged in the clinical context (personal and family history, symptoms and so on) and not on imaging grounds.

Minor comments:

Simple summary: summarize instead of summaries, evidence instead of evidences, may be necessary instead of is necessary.

lines 65-70: this mostly refers to RV VD description and comorbidities, rather than etiology.

line 80: or a instead of ore

lines 82-86: meaning unclear, the expression needs to be improved

Author Response

Reviewer #3

Comment # 1

In this systematic review, Pavasini et al. examine studies on various imaging techniques for ventricular outpouchings (diverticula and congenital aneurysms) and invaginations (recesses and crypts). Based on the more frequent occurrence of such imaging findings with CT/MRI, they propose the use of the latter on top of echocardiography for outpouching/invagination workup.

The objective/aim of this review is not clear, especially given the recent systematic review by Hojland et al. https://doi.org/10.1002/jcu.23155 As far as I can understand, the authors aim to summarize the data on outpouchings/invaginations from CT/MRI studies.

Reply # 1

We are sorry for being not precise about this point. The systematic review quoted by the reviewer is different from ours considering that as stated by the authors the main aim of Hojland et al. is to “ provide an overview of echocardiographic features of LV recess, cleft, diverticulum, pseudoaneurysms/aneurysms, and non- compaction based upon review of the literature as well as present some relevant clinical cases from our echocardiography labs”. As stated in introduction the main aim of our systematic review is to summaries the current evidences and the use of multimodality imaging in diagnosis, differential diagnosis and prognostic stratification regarding these rare disorders. We better underline this concept also in methods, considering that it is the use of multimodality imaging the distinctive figure compared to the manuscript of Hojland et al.

Modified text: Section Methods

The main aim of the systematic review is to summarize data regarding diagnosis, epidemiology and prognosis regarding ventricular outpouchings and invaginations as assessed by the use of multimodality imaging (namely echocardiography, CMR or CCT).

Comment # 2

As ventricular diverticulum definition is not universally accepted, for every study including such findings, the definition that the authors applied should be visible. 

Reply # 2

We thank the reviewer for this suggestion. We have added a column in Table 2, regarding definitions of structural abnormalities described in every dedicated study.

Modified text: Table 2

Comment # 3

When specific case series/cohorts are mentioned (for example lines 23-50), the ancestry-race of subjects should be included (caucasians, asians, africans…) as it may affect the findings.

Reply # 3

We thank the reviewer for this suggestion. We reviewed all the studies included in the systematic review, but none of them specified the ethnicity of recipients included.

Comment # 4

The authors conclude that outpouchings/invaginations are more frequently detected with CT/MRI than with echo, nevertheless they recommend echo as the first-line imaging modality. On the other hand, given the usually benign course of such findings, it is debatable both if a negative echo of adequate quality should lead to MRI and if a positive echo (for example indicative of one or two LV recesses or a typical aneurysm) should be followed by an MRI scan. Indications for cardiac CT should be restricted to non-availability of MRI. Thus, the need for additional imaging (on top of TTE) should be judged in the clinical context (personal and family history, symptoms and so on) and not on imaging grounds.

Reply # 4

We are sorry for being not enough precise about this point. We agree with the Reviewer that CMR and CCT should be used on top of echocardiography base on clinical context (family history of HCM or LVNC or other cardiomyopathies and symptoms) for diagnosis of structural abnormalities. As opposite in general population once after an echocardiogram there is the doubt of being faced with a structural anomaly, second-level cardiac imaging will have to be used, preferring CMR and, where not available, CCT for diagnostic confirmation. We better underline this concept in conclusions.

Modified text: Conclusions

In conclusion, in case of symptoms of suggestive clinics in patient with family history of cardiomyopathy (e.g. HCM or LVNC) CMR or CCT might be used on top of echocardiography for detection of outpouchings and invaginations. In general population multimodality imaging might be used in case of echocardiographic doubt of SA, trying to prefer CMR to CCT.

Comment #5

Minor comments:

 Simple summary: summarize instead of summaries, evidence instead of evidences, may be necessary instead of is necessary.

line 80: or a instead of ore

lines 82-86: meaning unclear, the expression needs to be improved

Reply # 5

We are sorry for typos. We have extensively reviewed the manuscript and amended typos.

Modified text: Section “simple summary”

For the differential diagnosis of these cardiac abnormalities multimodality imaging with a combined use of echocardiography, cardiac computed tomography (CCT) and cardiac magnetic resonance (CMR) may be necessary. The present systematic review aims to summarize the current evidence and the use of multimodality imaging in diagnosis, differential diagnosis and prognostic stratification regarding these rare disorders.

Modified text: page 3 line 79-80

Typical findings in CCTA for VD are protruding structures with a sac-like or tube-like shape and narrow orifice from the ventricular lumen

Modified text: page 3 line 82-86

Utility of evaluation of VA with CCT is more debated since it can provide information on coronary artery anatomy in order to exclude secondary causes of VA. CCT should be used for evaluation of morphology and tissue characteristics only when echocardiography is not conclusive and CMR unavailable [27].

Comment # 6

lines 65-70: this mostly refers to RV VD description and comorbidities, rather than etiology.

Reply #6

We thank the Reviewer for the comment. However considering the overall structure of the review and the absence of a section named “description and comorbidities” we believe this paragraph is in the most appropriate position (although the observations of the review are correct).

Round 2

Reviewer 3 Report

English editing is still needed.

Abstract:

"In conclusion both outpouchings and invaginations are rare entities: a definitive diagnosis may be aided by the use of combining multiple imaging techniques, and the treatment depends both on the lesion-specific risk of complications and on the potential association of some lesions with cardiomyopathy."

Author Response

Reviewer # 3

Comment # 1

English editing is still needed.

Abstract:

"In conclusion both outpouchings and invaginations are rare entities: a definitive diagnosis may be aided by the use of combining multiple imaging techniques, and the treatment depends both on the lesion-specific risk of complications and on the potential association of some lesions with cardiomyopathy."

Reply #1

We thank the Reviewer for the suggestion. We amended the text as suggested and we reviewed the full manuscript once again for English.

Modified text: abstract

In conclusion both outpouchings and invaginations are rare entities: a definitive diagnosis may be aided by the use of combining multiple imaging techniques, and the treatment depends both on the lesion-specific risk of complications and on the potential association of some lesions with cardiomyopathy

Modified text: English corrections throughout the manuscript.
